# Complex structure of cytochrome *c*–cytochrome *c* oxidase reveals a novel protein–protein interaction mode

Satoru Shimada[1], Kyoko Shinzawa-Itoh[1,*] (iD), Junpei Baba[1], Shimpei Aoe[1], Atsuhiro Shimada[1], Eiki Yamashita[2], Jiyoung Kang[1], Masaru Tateno[1], Shinya Yoshikawa[1] & Tomitake Tsukihara[1,2,3,**]

## Abstract

**Mitochondrial cytochrome *c* oxidase (CcO) transfers electrons from cytochrome *c* (Cyt.*c*) to $O_2$ to generate $H_2O$, a process coupled to proton pumping. To elucidate the mechanism of electron transfer, we determined the structure of the mammalian Cyt.*c*–CcO complex at 2.0-Å resolution and identified an electron transfer pathway from Cyt.*c* to CcO. The specific interaction between Cyt.*c* and CcO is stabilized by a few electrostatic interactions between side chains within a small contact surface area. Between the two proteins are three water layers with a long inter-molecular span, one of which lies between the other two layers without significant direct interaction with either protein. Cyt.*c* undergoes large structural fluctuations, using the interacting regions with CcO as a fulcrum. These features of the protein–protein interaction at the docking interface represent the first known example of a new class of protein–protein interaction, which we term "soft and specific". This interaction is likely to contribute to the rapid association/dissociation of the Cyt.*c*–CcO complex, which facilitates the sequential supply of four electrons for the $O_2$ reduction reaction.**

**Keywords** cytochrome *c*; cytochrome *c* oxidase; electron transfer complex; protein–protein interaction; X-ray crystallography

**Subject Categories** Membrane & Intracellular Transport; Metabolism; Structural Biology

**The EMBO Journal (2017) 36: 291–300**

See also: **JA Lyons & P Nissen** (February 2017)

## Introduction

Cytochrome *c* oxidase (CcO) is a typical $aa_3$-type CcO, in which electrons are transferred to an active site consisting of heme $a_3$ and $Cu_B$ from $Cu_A$ via heme *a*. CcO initially accepts electrons from cytochrome *c* (Cyt.*c*) to reduce a dioxygen molecule (Ferguson-Miller & Babcock, 1996; Yoshikawa & Shimada, 2015). Electron transfer (ET) from $Cu_A$ to the $O_2$ reduction center is coupled to proton pumping across the membrane.

Extensive steady-state kinetic analyses of oxidation of ferro-Cyt.*c* by CcO have revealed two Cyt.*c*-binding sites, both of which are actively involved in catalytic turnover (Ferguson-Miller *et al*, 1976). Speck *et al* (1984) proposed a single-catalytic site model in which one binding site is the catalytic site through which electrons are transferred, whereas the other controls ET in the catalytic site. The amino-acid residues on the Cyt.*c* surface that interact with CcO were examined by chemically modifying basic residues of Cyt.*c* and observing the effect on CcO activity; these experiments revealed the critical involvement of basic residues on the Cyt.*c* surface (Ferguson-Miller *et al*, 1978; Osheroff *et al*, 1980). The residues that interact with CcO have been investigated more extensively by NMR studies (Sakamoto *et al*, 2011), which revealed that hydrophobic residues on the surface of Cyt.*c* make major contributions to complex formation, whereas the charged residues near the hydrophobic core refine the orientation of Cyt.*c* to precisely control ET. However, as noted above, analyses of the mechanism of ET between Cyt.*c* and CcO have been largely restricted to the Cyt.*c* side. With the exception of docking simulation analyses (Roberts & Pique, 1999; Sato *et al*, 2016), essentially no experimental information is available regarding the Cyt.*c*-binding surface of CcO.

Although a significant amounts of data have accumulated regarding ET from Cyt.*c* to CcO (Speck *et al*, 1984; Sakamoto *et al*, 2011), and the X-ray structures of mammalian CcO (PDB 5B1A) and Cyt.*c* (Bushnell *et al*, 1990; De March *et al*, 2014) have been determined at high resolution, the underlying mechanism of ET remains incompletely understood. A crystal structure of the complex of CcO and Cyt.*c* would be invaluable for mechanistic studies, but to date no structure of a Cyt.*c*–CcO complex has been determined other than that of $caa_3$-type CcO from *Thermus thermophilus* (Lyons *et al*,

1  Picobiology Institute, Graduate School of Life Science, University of Hyogo, Akoh, Hyogo, Japan
2  Institute for Protein Research, Osaka University, Suita, Osaka, Japan
3  JST, CREST, Kawaguchi, Saitama, Japan
   *Corresponding author. Tel: +81 791 58 0191; E-mail: shinzawa@sci.u-hyogo.ac.jp
   **Corresponding author. Tel: +81 791 58 0342, +81 6 6879 4331; E-mail: tsuki@protein.osaka-u.ac.jp

2012), which has a covalently tethered cytochrome *c* domain. Thus, it remains unclear whether this fused Cyt.*c* has functions analogous to those of the Cyt.*c* molecules that participate in catalytic turnover in the eukaryotic C*c*O system.

Two-dimensional (2D) crystals of the mammalian Cyt.*c*–C*c*O complex were prepared at higher pH (7.4–9.0) with both proteins in the oxidized state (Osuda *et al*, 2016), but these 2D crystals could not provide a structure of sufficient resolution to allow a detailed analysis of the interactions between these proteins. Therefore, in this study, we optimized the three-dimensional (3D) crystallization conditions for ferri-Cyt.*c* and oxidized C*c*O at high pH and solved the X-ray structure of the complex at 2.0-Å resolution. The results revealed a novel mode of protein–protein interaction mediated by three water layers.

## Results and Discussion

### Crystallization of the Cyt.*c*–C*c*O complex

Previously, bovine C*c*O stabilized with *n*-decyl-β-D-maltoside (DM) was crystallized at a pH ≤ 6.8 and analyzed at the atomic level (Tsukihara *et al*, 1995, 1996). However, no 3D crystallization trial of the Cyt.*c*–C*c*O complex has been successful under crystallization conditions similar to those used for C*c*O at low pH. Therefore, we performed co-crystallization of Cyt.*c* and C*c*O at pH 8.0 under the same conditions used for 2D crystallization of the Cyt.*c*–C*c*O complex (Osuda *et al*, 2016). C*c*O purified from bovine heart was solubilized with DM and fluorinated octyl-maltoside (FOM), followed by addition of horse Cyt.*c* at a Cyt.*c*/C*c*O molar ratio of 1.2. The Cyt.*c*–C*c*O complex was then co-crystallized by the batch-wise method at 277 K (Appendix Fig S1A). Absorption spectral analysis indicated that the resultant crystals contained both Cyt.*c* and C*c*O (Appendix Fig S1B). The crystals were soaked in a crystallization solution both containing 50 μM Cyt.*c* and gradually increasing concentrations of the cryo-protectant ethylene glycol (EG; 40% at the final step), and then frozen in a cryo-nitrogen stream at 100 K. The addition of 50 μM Cyt.*c* prevented the crystal from deterioration due to release of Cyt.*c* molecules from the complex during soaking. These crystals diffracted X-rays to a resolution of 1.8 Å (Appendix Fig S1C and D). Statistics of the intensity data and structure refinement at 2.0-Å resolution are provided in Table 1.

### Structure determination and overall structure of the Cyt.*c*–C*c*O complex

Initial phases were determined by the molecular replacement (MR) method (Rossmann & Blow, 1962) using C*c*O, and Cyt.*c* molecules were located using the $F_o$–$F_c$ difference map and anomalous difference map (Appendix Fig S2A and B). Structure refinement at 2.0-Å resolution converged well: $R$/$R_{free}$ = 0.167/0.207; r.m.s.d of bond lengths = 0.023 Å; r.m.s.d. of bond angles = 2.0° (Table 1). The ($2F_o$–$F_c$) electron-density map for the interface of Cyt.*c* and C*c*O clearly shows electron densities of side chains interacting with their counterpart proteins (Appendix Fig S2C).

The asymmetric unit of the monoclinic lattice contains a dimer consisting of two complexes of C*c*O and Cyt.*c*. The dimeric structure of C*c*O of the Cyt.*c*–C*c*O complex is almost identical to that of C*c*O

**Table 1. Data collection and refinement statistics of Cyt.*c*–C*c*O complex crystals.**

| Data collection | |
| --- | --- |
| Space group | $P2_1$ |
| Cell dimensions | |
| *a*, *b*, *c* (Å) | 113.3, 183.9, 148.9 |
| β (°) | 102.1 |
| Resolution (Å) | 50–2.0 (2.02–2.00) |
| Observed reflections | 1,960,373 |
| Independent reflections | 397,399 (9,885) |
| Averaged redundancy | 4.9 (3.7) |
| $I/\sigma(I)$ | 17.0 (1.2) |
| Completeness (%) | 99.5 (99.5) |
| $R_{merge}$ | 0.097 (> 1.0) |
| $R_{p.i.m.}$ | 0.044 (0.599) |
| $CC_{1/2}$ | 0.903 (0.623) |
| **Refinement** | |
| Resolution (Å) | 40–2.0 |
| No. reflections (all/free) | 377,338/19,990 |
| $R^{||}$/$R_{free}$ | 0.167/0.207 |
| No. atoms | |
| C*c*O | 28,969 |
| Cyt.*c* | 1,738 |
| Others | 4,348 |
| Averaged $B$-factors (Å$^2$) | |
| C*c*O A mol. | 39.7 |
| C*c*O B mol. | 47.3 |
| Cyt.*c* A mol. | 88.4 |
| Cyt.*c* B mol. | 111.6 |
| Others | 67.1 |
| R.m.s.d. bond lengths (Å) | 0.023 |
| R.m.s.d. bond angles (°) | 2.0 |

R.m.s.d., Root-mean-square deviation.
Values in parentheses are for highest-resolution shell.

crystallized in an orthorhombic lattice (Tsukihara *et al*, 1996). As in the C*c*O orthorhombic crystal (Tomizaki *et al*, 1999), one of the two C*c*O molecules in the asymmetric unit had a lower *B*-factor than the other, by about 7 Å$^2$, and no significant structural difference was detected between the two complexes. Furthermore, structure refinement was performed under non-crystallographic symmetry restraint between two C*c*O molecules; therefore, we focused our structural descriptions on this complex.

Cyt.*c* is localized on the positive side of a concave surface of C*c*O (Figs 1 and EV1). Although *B*-factors of Cyt.*c* were significantly higher than those of C*c*O, all the side chains except for Lys[25] were located in the positive density of ($2F_o$–$F_c$) map. Consistent with the results of studies in which Cyt.*c* was chemically modified at lysyl residues, C*c*O interacts with the front surface of Cyt.*c* in a region that includes the exposed heme edge of Cyt.*c* (Ferguson-Miller *et al*, 1978; Osheroff *et al*, 1980). By contrast, in *caa*$_3$-type C*c*O, the

**Figure 1.  Overall structure of the Cyt.c–CcO complex at 2.0-Å resolution.**

Cyt.c, subunit II of CcO, and other subunits of CcO are shown in red, light blue, and green, respectively.

A, B   Ribbon drawing of the Cyt.c–CcO complex, viewed from the trans-membrane surface (A) and the positive side (B).

C   Close-up view of the interface between Cyt.c and the subunit II of CcO, shown as surface representation.

propionate side of heme group of Cyt.c faces $Cu_A$ (Lyons *et al*, 2012). CcO interacts with Cyt.c mainly via subunit II, with 94% of the contact surface of CcO with Cyt.c belonging to subunit II, and 5 and 1% of it belonging to subunits VIb and I, respectively.

The closest inter-atomic distance between Cyt.c and any Cyt.c–CcO complex related by crystallographic symmetry is 6.9 Å. Because Cyt.c does not interact directly with any symmetry-related Cyt.c–CcO complexes (Appendix Fig S3), molecular packing in the crystal does not perturb the structure of Cyt.c in the Cyt.c–CcO complex. The CcO and Cyt.c structures in the complex superpose well with the previously determined structures of the individual proteins, as shown in Appendix Fig S4. At the current resolution, docking of CcO and Cyt.c results in no significant structural changes in the main chains. In Appendix Fig S5, phospholipids are depicted (sticks) along with the $C_\alpha$ traces of the complex (ribbons). All phospholipids detected in the crystal structure of CcO (PDB 5B1A), but no additional lipids, are present in the Cyt.c–CcO complex. None of these phospholipids are localized near the Cyt.c-binding site; therefore, Cyt.c does not interact with phospholipids in the crystal of the complex.

**A possible electron transfer pathway from heme c to $Cu_A$**

The concave surface consists of subunit II, which contains $Cu_A$, the first loading site for electrons transferred from Cyt.c. The distance between the iron atom of heme c and the copper atom of $Cu_A$ is 23.0 Å. The dominant ET pathway from the heme c iron to $Cu_A$ of CcO was explored using the *Pathways* plugin for VMD (Humphrey *et al*, 1996; Balabin *et al*, 2012). The calculations suggest that the most probable ET pathway, as shown in Figs 2 and EV2, proceeds through the iron atom of heme c (Cyt.c)-Cys[14] (Cyt.c)-Lys[13] (Cyt.c)-Tyr[105] (subunit II of CcO)-Met[207] (subunit II of CcO)-$Cu_A$. This ET pathway contains two short through-space jumps: one from the $N_\zeta$ atom of Lys[13] (Cyt.c) to the $C_\zeta$ atom of Tyr[105] (subunit II of CcO) (a distance of 3.6 Å), and the other from the N atom of the main chain of Tyr[105] (subunit II of CcO) to the $S_\delta$ atom of Met[207] (subunit II of CcO) (a distance of 4.0 Å). The running distance along the pathway is 41.9 Å. Chemical modification of Lys[13] of Cyt.c induces drastic inhibition on ET activity (Ferguson-Miller *et al*, 1978; Osheroff *et al*, 1980). All vertebrate Cyt.c proteins contain Lys at the 13[th] residue and Cys at the 14[th] residue (Appendix Fig S6A); in addition, Tyr[105]

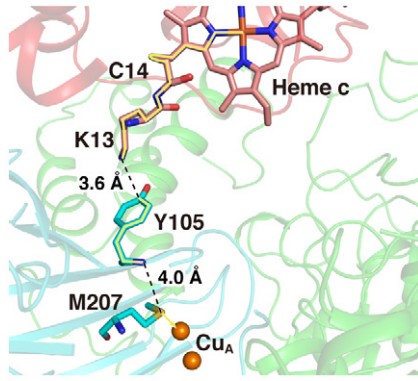

**Figure 2.  Electron transfer pathway in the Cyt.c–CcO complex.**

The possible electron transfer (ET) pathway from the heme c iron to $Cu_A$ was explored using the *Pathways* plugin for VMD (Humphrey *et al*, 1996; Balabin *et al*, 2012). Through-bond processes and through-space jumps are represented by yellow solid lines and black dashed lines, respectively. The heme c group and amino-acid residues of Cyt.c molecule are shown as pink sticks. The amino-acid residues of subunit II of CcO are shown as cyan sticks. Oxygen, nitrogen, and sulfur atoms are shown in red, blue, and yellow, respectively. Orange spheres indicate $Cu_A$. Protein structures are drawn by transparent ribbons in the same colors as those of Fig 1.

and Met[207] of CcO subunit II are conserved among vertebrates (Appendix Fig S6B). These observations strongly suggest that the Cyt.c-binding site in the Cyt.c–CcO complex structure is the catalytic binding site of Cyt.c through which electrons are transferred. Compared with the pathway in $caa_3$-type CcO, only the ET from Met[207] (subunit II of CcO) to $Cu_A$ is conserved (Fig EV2) (Lyons *et al*, 2012).

It has been proposed that Trp[121] in *Paracoccus denitrificans* CcO (Trp[104] in subunit II of bovine CcO) is the electron entry site from Cyt.c, based on W121Q mutation (Witt *et al*, 1998). The proposal was supported by some docking simulations (Roberts & Pique, 1999; Drosou *et al*, 2002). However, this mutation is likely to greatly influence the redox potential of $Cu_A$, because the side chain of Gln[121] in the W121Q mutant is predicted to make hydrogen bonds with both the $S_\delta$ atom of Met[227] and the $S_\gamma$ atom of Cys[220], which coordinate to copper ions of $Cu_A$ (Appendix Fig S7B). Furthermore, the present X-ray structure of the complex shows that Trp[104] does

not interact tightly with Cyt.*c*. There is a large gap between the protein surface around Trp[104] and Cyt.*c*, in which Trp[104] is separated from the closest atom of heme *c* by 9.0 Å (Appendix Fig S7A). Thus, the possible electron transfer pathway identified in the present X-ray structural analyses suggests a significantly more facile electron transfer than the one through the structure including Trp[104] (Appendix Fig S7A).

## Catalytic binding sites

Speck *et al* (Speck *et al*, 1984) proposed a single-catalytic site model including a catalytic site and a non-catalytic regulatory site on C*c*O for Cyt.*c* to interpret the steady-state kinetic results indicating two different Michelis–Menten kinetics, without giving any experimental confirmation. In other words, no experimental result has disproven the two-catalytic site model (Ferguson-Miller *et al*, 1976). Following the Speck's definition, the above structure strongly suggests the catalytic binding site, since the Cyt.*c*–C*c*O complex shows a facile electron transfer pathway from heme *c* to Cu$_A$. However, following the two-catalytic site model, this retains both possibilities of the first and the second catalytic sites.

The positive side of the concave surface of C*c*O is negatively charged, whereas the surface area around the exposed heme edge of Cyt.*c* is positively charged (Fig 3A and B). Prominent intermolecular interactions in this region include six hydrogen bonds or salt bridges between C*c*O and Cyt.*c* (Fig 3C and Table 2). Lys[8] (Cyt.*c*) interacts with Asp[139] of subunit II (C*c*O) via a salt bridge, Gln[12] (Cyt.*c*) forms hydrogen bonds with Asp[139] (C*c*O subunit II), Lys[13] (Cyt.*c*) forms hydrogen bonds with Tyr[105] and Tyr[121] (C*c*O subunit II) and a salt bridge with Asp[119] (C*c*O subunit II), and Lys[87] (Cyt.*c*) forms hydrogen bonds with Ser[117] (C*c*O subunit II). These four interacting residues of Cyt.*c* are restricted to the molecular surface near the exposed heme edge (Fig 3B). On the basis of chemical modification and kinetic studies (Ferguson-Miller *et al*, 1978), three lysine residues, Lys[8], Lys[13], and Lys[87], were predicted to interact with C*c*O. Recent site-directed mutagenesis and kinetics studies of Cyt.*c* indicated that the ET activities of K13L, K86L/K87L, and K7L/K8L mutants are significantly lower than that of the wild-type protein (Sato *et al*, 2016). The side chains of Lys[8], Gln[12], Lys[13], and Lys[87] of Cyt.*c*, as well as the side chains of Tyr[105], Asp[119], Ser[117], Tyr[121], and Asp[139] of C*c*O subunit II, provide the physiological electron transfer complex, not an encounter complex under nonphysiological conditions.

A previous NMR study (Sakamoto *et al*, 2011) detected structural changes in several hydrophobic amino-acid residues of Cyt.*c* upon the docking of two proteins, and the authors of that study concluded that Cyt.*c* interacted with C*c*O via its non-polar surface surrounding the heme cleft, as in the cytochrome *bc*$_1$ complex (Cyt.*bc*$_1$)–Cyt.*c* (Lange & Hunte, 2002) and Cyt.*c*–cytochrome *c* peroxidase (C*c*P) complexes (Jasion *et al*, 2012). By contrast, our crystal structure of the Cyt.*c*–C*c*O complex has no inter-molecular interactions between hydrophobic amino acids with an inter-atomic distance < 5 Å. This is likely because NMR spectroscopy sensitively detected a small structural change undetectable by X-ray, mediated by an interaction between the residues of Cyt.*c* and C*c*O via water molecules present between the two proteins.

The ionic interaction between Lys[13] (Cyt.*c*) and Asp[119] (C*c*O) was predicted by a docking simulation (Roberts & Pique, 1999), and

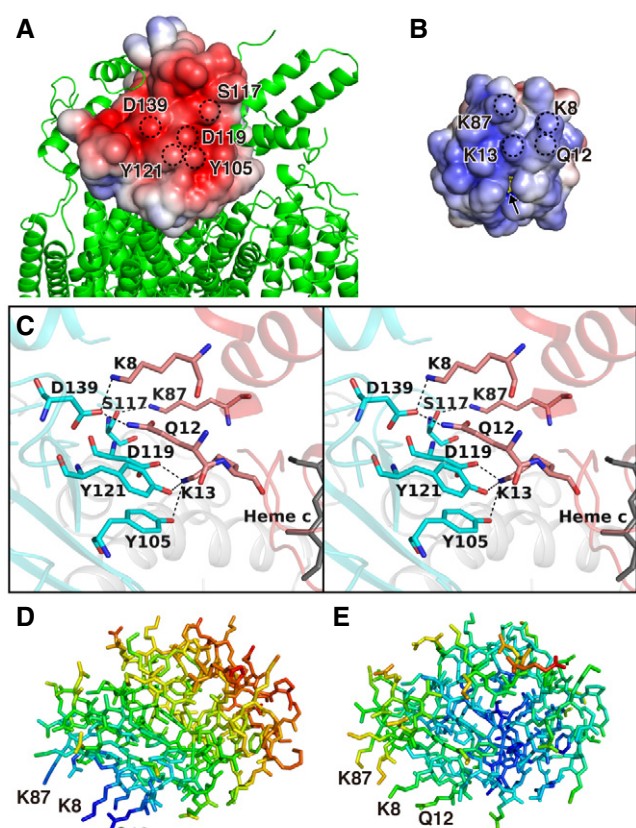

**Figure 3. Cyt.*c*–C*c*O interaction.**

A, B Open-book view of electrostatic potentials of the interaction surfaces of subunit II (residues 91–227) of C*c*O (A) and Cyt.*c* (B) in the Cyt.*c*–C*c*O complex. Electrostatic potentials were calculated separately using the program APBS (Baker *et al*, 2001). The displayed potentials range from −5 (red) to 5 (blue) $kТe^{-1}$. Heme *c* (yellow sticks), slightly exposed to the surface, is indicated by an arrow. Dotted circles indicate sites of amino-acid residues, indicated by single-letter notation with residue number.

C Close-up view of the interaction site, shown as a stereoscopic pair. Amino-acid residues involved in the interaction between Cyt.*c* and C*c*O are represented by pink (Cyt.*c*) and cyan sticks (subunit II of C*c*O). Oxygen and nitrogen atoms are shown in red and blue, respectively. Hydrogen bonds and salt bridges are shown as dashed lines. Amino-acid residues are indicated by single-letter notation with residue number.

D, E Comparison of variation in *B*-factors in the Cyt.*c* molecule. Stick representation of Cyt.*c* molecule in the Cyt.*c*–C*c*O complex (D) and free Cyt.*c* (PDB 1HRC) (E); colors represent *B*-factor ranging from 37.7 Å$^2$ (blue) to 136.0 Å$^2$ (red) in (D) and 5.8 Å$^2$ (blue) to 77.4 Å$^2$ (red) in (E).

**Table 2. Protein–protein distances between Cyt.*c* and C*c*O.**

| Cyt.*c* | C*c*O (subunit II) | Distances (Å) |
|---|---|---|
| Heme *c* Fe | Cu$_A$ (CU1) | 23.0 |
| Lys8 N$_\zeta$ | Asp139 O$_{\delta 2}$ | 2.7 |
| Gln12 N$_{\varepsilon 2}$ | Asp139 O$_{\delta 2}$ | 2.9 |
| Lys13 N$_\zeta$ | Tyr105 O$_\eta$ | 3.3 |
| Lys13 N$_\zeta$ | Asp119 O$_{\delta 2}$ | 2.6 |
| Lys13 N$_\zeta$ | Tyr121 O$_\eta$ | 3.1 |
| Lys87 N$_\zeta$ | Ser117 O | 2.9 |

 

another docking simulation assigned Lys[8], Lys[13], and Lys[87] of Cyt.*c* as residues interacting with C*c*O (Sato *et al*, 2016), as observed in this study. However, inconsistencies remain between the X-ray structure and the simulated structures of Cyt.*c*–C*c*O complex. The former simulation predicted that Lys[72] (Cyt.*c*), which is distant from C*c*O in the complex structure (Appendix Fig S8), interacts with Gln[103] and Asp[158] of C*c*O subunit II. The most probable structure from the latter simulation indicated that the subunit I of C*c*O had a larger contact surface area with Cyt.*c* than subunit II of C*c*O, whereas in our structure most of the contact surface of C*c*O with Cyt.*c* belongs to subunit II. These inconsistencies likely arose because water molecules are present between Cyt.*c* and C*c*O, but bulk waters were removed from the surfaces of both proteins in the docking simulations.

Amino-acid residues included in the catalytic binding were assigned based on the Cyt.*c*–C*c*O complex structure equilibrated in a solution in which the enzyme exerts its normal catalytic activity (Yonetani & Ray, 1965). The interactions between Cyt.*c* and C*c*O elucidated by this crystallographic study are consistent with those revealed for the enzyme–substrate complex under turnover conditions by previous experimental studies involving chemical modifications and kinetics (Ferguson-Miller *et al*, 1978) or solution NMR and kinetics for complexes containing wild-type and mutant Cyt.*c* proteins (Sakamoto *et al*, 2011; Sato *et al*, 2016).

### Novel protein–protein interaction scheme

Cyt.*c* donates electrons to C*c*O and (C*c*P) and accepts electrons from (Cyt.*bc*$_1$). We compared the interaction scheme of the Cyt.*c*–C*c*O complex with those of the Cyt.*c*–C*c*P (Pelletier & Kraut, 1992; Jasion *et al*, 2012) and the Cyt.*bc*$_1$–Cyt.*c* complex (Lange & Hunte, 2002; Solmaz & Hunte, 2008). The shortest distance between two C$_\alpha$ atoms of Cyt.*c* and C*c*O is 8.2 Å. By contrast, the shortest distances in the Cyt.*c*–C*c*P (PDB 4GED) and Cyt.*bc*$_1$–Cyt.*c* (PDB 3CX5) complexes are much shorter, 5.3 and 5.6 Å, respectively. Thus, Cyt.*c* in the Cyt.*c*–C*c*O complex is farther from C*c*O than it is from C*c*P and Cyt.*bc*$_1$ in the corresponding complexes. Ahmed *et al* (2011) compiled 179 X-ray structures of protein–protein complexes from the RSCB Protein Data Bank (Berman *et al*, 2000). The intermolecular C$_\alpha$ distances of these 179 structures were calculated, and the distribution of the shortest distance in each complex is illustrated in Appendix Fig S9. Notably, the shortest distance in the Cyt.*c*–C*c*O complex, 8.2 Å, falls well outside the distribution. Furthermore, the contact surface areas for three ET complexes were calculated by removing surface water molecules. The area of the Cyt.*c*–C*c*O complex (222.8 Å$^2$) is approximately one-third that of the Cyt.*c*–C*c*P complex (615.2 Å$^2$), and less than one-fourth that of Cyt.*bc*$_1$–Cyt.*c* (1008.7 Å$^2$). Thus, fewer direct protein–protein interactions are involved in formation of the Cyt.*c*–C*c*O complex than either of the other two complexes. No direct interaction (< 5.0 Å) between hydrophobic residues was detected in the Cyt.*c*–C*c*O complex, whereas the other two complexes have several non-polar groups involved in their inter-molecular interactions.

The water molecules within 7 Å of both proteins of Cyt.*c* and C*c*O fall into three categories, as noted by Ahmed *et al* (2011): bridging waters that interact with both proteins; non-bridging waters that interact with one but not both proteins; and non-interacting waters that are more than 3.5 Å from both proteins. In this study, interactions between waters and proteins atoms were assigned based on a distance of < 3.5 Å between the water oxygen atom and the nearest atom of the protein. As shown in Fig 4 and Appendix Table S1, more water molecules are present between Cyt.*c* and C*c*O than between Cyt.*c* and C*c*P or Cyt.*bc*$_1$ and Cyt.*c*, and there are a total of 14 non-interacting water molecules in the Cyt.*c*–C*c*O complex. By contrast, the Cyt.*c*–C*c*P and Cyt.*bc*$_1$–Cyt.*c* complexes each have only four and two non-interacting water molecules, respectively. Almost the same numbers of bridging waters are in three ET complexes. Extremely fewer non-bridging waters are at Cyt.*c* in the both complexes of Cyt.*bc*$_1$–Cyt.*c* and Cyt.*c*–C*c*P than Cyt.*c*–C*c*O complex (Fig 4; Appendix Table S1). Hydrophobic

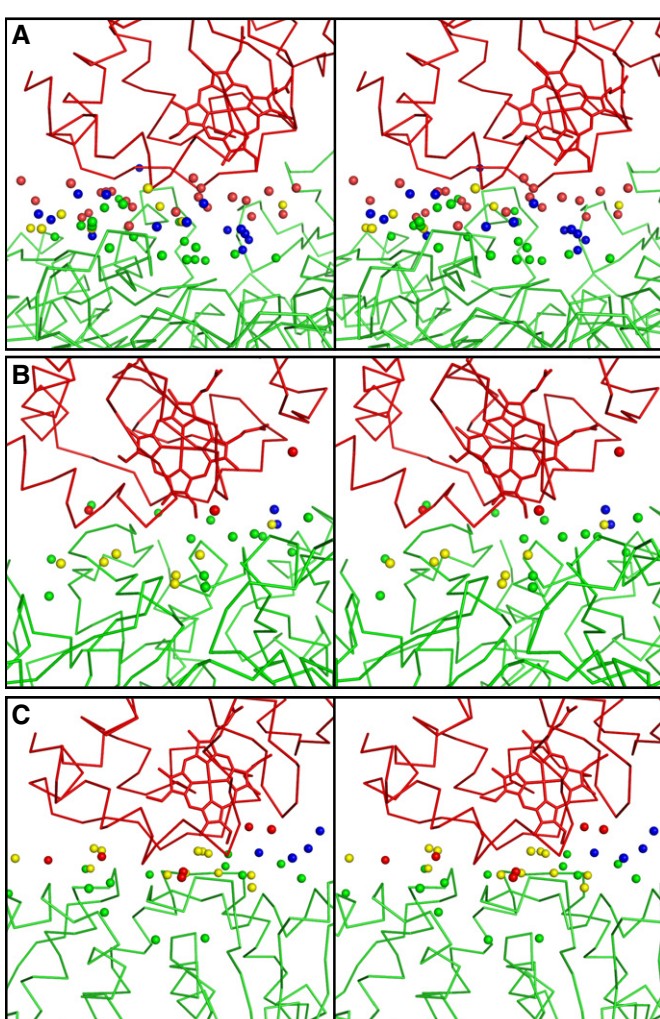

**Figure 4.   Comparison of the distribution of water molecules between Cyt.*c* and each redox partner.**

A–C   Stereo views of water molecules between proteins in Cyt.*c*–C*c*O (A), Cyt.*bc*$_1$–Cyt.*c* (PDB 3CX5) (B), and Cyt.*c*–C*c*P complexes (PDB ID 4GED) (C). Bridging and non-interacting waters are shown in yellow and blue, respectively. Non-bridging waters interacting with Cyt.*c* are red spheres, and that interacting with redox partners are shown in green. Cyt.*c* and the redox partners are shown by wire models in red and green, respectively.

residues of Cyt.*c* in the Cyt.*bc*$_1$–Cyt.*c* complex and the Cyt.*c*–CcP complex likely contact directly with their counterpart and remove waters in part from the surface of their Cyt.*c*.

Any water in the Cyt.*c*–CcO complex has at least one hydrogen bond with a protein atom or a water molecule. Each of the bridging and non-bridging waters in the Cyt.*c*–CcO complex interacts, on average, with three polar atoms or waters and one non-polar atom (Appendix Table S2). The waters at CcO interact prominently with Asp, whereas those at Cyt.*c* interact mainly with Lys and Gln (Appendix Table S3). At least two water molecules closely contact a non-interacting water molecule. These water molecules construct a hydrogen bond network between Cyt.*c* and CcO (Fig EV3). The averaged *B*-factor of 64 waters is 62.7 Å$^2$, which is between that of CcO (37.9 Å$^2$) and that of Cyt.*c* (88.2 Å$^2$).

Out of 19 non-bridging waters at CcO in the complex, 14 are located at almost the same sites in the CcO crystal (PDB 5B1A), four are in slightly shifted positions, and one water is not assigned in the CcO crystal. Out of 23 non-bridging waters at Cyt.*c* in the complex, only five waters are present in the Cyt.*c* crystal structure (PDB 1HRC). This is probably because the interacting sites of Cyt.*c* in the Cyt.*c*–CcO complex are involved in the tight contacts of crystal packing in the Cyt.*c* crystal, which removes waters from the molecular surface upon crystallization. Because the protein volumes of Cyt.*c*–CcO and CcO crystals are ~30% of their unit cell volume, significantly lower than that of Cyt.*c* crystal, more than 40%, non-bridging water sites are common to the Cyt.*c*–CcO and CcO crystals.

When Cyt.*c* docks with CcO, both proteins preserve their main chain folds, and retain water molecules on their surfaces, and they interact with each other via the long arms of side chains (Fig 5A). On the other hand, the docking of Cyt.*c* and CcP or Cyt.*bc*$_1$ leads to the exclusion of water molecules from the surface of each protein (Fig 5B). The chemical shift-perturbed residues of ferri-Cyt.*c* associated with the binding of CcO (Sakamoto *et al*, 2011) are not affected by direct protein–protein interactions, but are influenced by indirect interactions via the water layers in the crystal structure.

The main chain folds of CcO and Cyt.*c* in Cyt.*c*–CcO complex are almost identical to those of the corresponding crystals, with C$_\alpha$ r.m.s.d. values of 0.47 and 0.41 Å, respectively. All the CcO side chain structures in the region interacting with Cyt.*c* are similar to those of the CcO crystal except for Asn$^{203}$(subunit II of CcO), where the two structures are different from each other by a −90° rotation angle around the C$_\beta$–C$_\gamma$ bond. By contrast, several side chains of Cyt.*c* in the interacting region have different orientations between the crystals of Cyt.*c*–CcO and Cyt.*c*, probably because of packing effects in the Cyt.*c* crystal, as noted above for non-bridging waters at Cyt.*c*.

The *B*-factors of the side chain atoms of N$_\zeta$ (Lys$^8$), N$_\epsilon$ (Gln$^{12}$), N$_\zeta$ (Lys$^{13}$), and N$_\zeta$ (Lys$^{87}$) of Cyt.*c*, which interact with residues of CcO, are 37.7, 39.3, 41.7, and 48.7 Å$^2$, respectively, significantly lower than the average *B*-factor of Cyt.*c* (88.2 Å$^2$) and as low as that of the extracellular domain (residues 91–227) of CcO subunit II (35.2 Å$^2$). By contrast, the *B*-factors of Cyt.*c* atoms increase with distance from these CcO-interacting residues (Fig 3D). Cyt.*c* in the Cyt.*c*–CcO complex undergoes a large fluctuation, using the regions interacting with CcO as a fulcrum. The variation in the *B*-factors in Cyt.*c* molecule of the Cyt.*c*–CcO complex is another unique feature of the ET complex. However, horse ferri- and ferro-Cyt.*c* and the Cyt.*c* molecules in the Cyt.*c*–CcP and Cyt.*bc*$_1$–Cyt.*c* complexes exhibit low

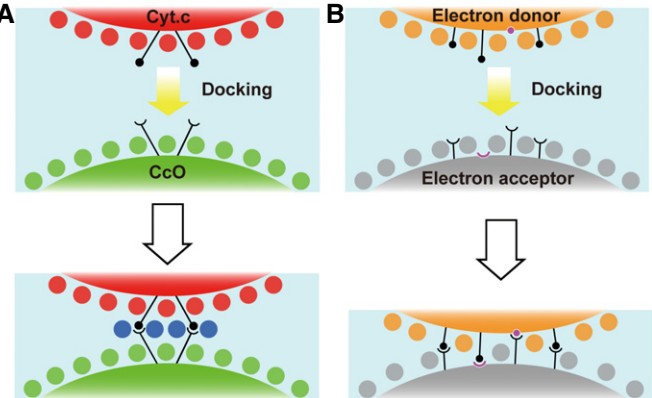

**Figure 5. Schematic representation of the distribution of water molecules at the protein–protein interaction site.**

A   In the Cyt.*c*–CcO complex system, water molecules on the surfaces of each protein are preserved to form three layers upon docking, but each protein specifically interacts via the long arms of side chains. Red and green circles represent water molecules on the surfaces of proteins of the corresponding color; blue circles represent water molecules belonging to the non-interacting water. Black circles and semicircles represent side chains of residues involved in protein–protein interactions.

B   In other ET complex systems, electron donor and acceptor proteins form an ET complex by excluding water molecules from the surface of each protein. Orange and gray circles represent water molecules on the surfaces of proteins of the corresponding color. Magenta circles and semicircles represent the main chains of residues involved in protein–protein interactions. Black circles and semicircles represent side chains of residues involved in protein–protein interactions.

*B*-factors at the heme *c* group (Figs 3E and EV4). The high flexibility of Cyt.*c* in the Cyt.*c*–CcO complex is likely to compensate for the entropy loss caused by introduction of more waters upon docking.

The inter-molecular interaction between Cyt.*c* and CcO is characterized by mutual recognition mediated by a few long arms of hydrophilic amino acids, small contact surface, a long span between the two proteins, the presence of three water layers between the two proteins, and a large fluctuation of Cyt.*c* that uses the regions

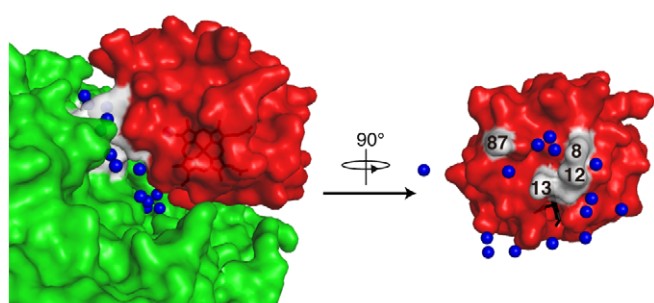

**Figure 6. Locations of four interacting residues in relation to Cyt.*c* and water molecules that engage in no direct interactions with either protein in the Cyt.*c* molecule.**

Cyt.*c* (red) and CcO (green) are shown as surface representations. The heme *c* group shown as black sticks is slightly exposed to the surface. Blue spheres represent water molecules belonging to the non-interacting water. Amino-acid residues involved in interactions between Cyt.*c* and CcO are shown in white, and figures indicate their residue numbers.

that interact with CcO as a fulcrum. The non-interacting water molecules in the Cyt.*c*–CcO complex exist in vacant spaces around the interacting amino-acid residues of both proteins (Fig 6) and closely contact with water molecules, thus providing hydrogen bond network between Cyt.*c* and CcO (Fig EV3). This novel mode of protein–protein interaction, which we term "soft and specific contact", is not observed in other ET complexes (Table 3) (Shen *et al*, 1994; Morales *et al*, 2000; Müller *et al*, 2001; Axelrod *et al*,

**Table 3.  Shortest distance between $C_\alpha$ atoms and contact surface areas of ET complexes.**

| PDB | Distance (Å) | CSA[a] (Å²) | Resolution (Å) | References |
|---|---|---|---|---|
| 5IY5[b] | 8.24 | 222.8 | 2.00 | This study |
| 4GED[c] | 5.28 | 615.2 | 1.84 | Jasion *et al* (2012) |
| 3CX5[d] | 5.57 | 1008.7 | 1.90 | Solmaz and Hunte (2008) |
| 1E6E | 4.71 | 1157.8 | 2.30 | Müller *et al* (2001) |
| 1EWY | 4.10 | 1361.1 | 2.38 | Morales *et al* (2000) |
| 1L9B | 5.30 | 722.8 | 2.40 | Axelrod *et al* (2002) |
| 1OAO | 4.40 | 2248.3 | 1.90 | Darnault *et al* (2003) |
| 2DE5 | 4.48 | 932.2 | 1.90 | Ashikawa *et al* (2006) |
| 2GC4 | 4.36 | 425.2 | 1.90 | Shen *et al* (1994) |
| 2IAA | 4.91 | 557.1 | 1.95 | Sukummar *et al* (2006) |
| 2PU9 | 4.67 | 952.0 | 1.65 | Dai *et al* (2007) |
| 2PVG | 5.53 | 636.6 | 2.40 | Dai *et al* (2007) |
| 2V3B | 4.76 | 657.1 | 2.45 | Hagelueken *et al* (2007) |
| 2YVJ | 5.11 | 690.9 | 1.90 | Senda *et al* (2007) |
| 2ZON | 4.06 | 533.6 | 1.70 | Nojiri *et al* (2009) |
| 3QFA | 4.81 | 631.7 | 2.20 | Fritz-Wolf *et al* (2011) |
| 3W9C | 5.36 | 595.9 | 2.50 | Hiruma *et al* (2013) |
| 4FA9 | 4.05 | 1433.8 | 2.09 | Yukl *et al* (2013) |
| 4PIB | 4.29 | 798.7 | 2.05 | Acheson *et al* (2014) |
| 4PW9 | 4.41 | 739.1 | 2.49 | McGrath *et al* (2015) |

[a]Contact surface area was calculated with the program AREAIMOL (Lee & Richards, 1971) in CCP4.
[b]Cyt.*c*–CcO complex.
[c]Cyt.*bc*₁–Cyt.*c* complex.
[d]Cyt.*c*–CcP complex.

2002; Darnault *et al*, 2003; Ashikawa *et al*, 2006; Sukummar *et al*, 2006; Dai *et al*, 2007; Hagelueken *et al*, 2007; Senda *et al*, 2007; Nojiri *et al*, 2009; Fritz-Wolf *et al*, 2011; Hiruma *et al*, 2013; Yukl *et al*, 2013; Acheson *et al*, 2014; McGrath *et al*, 2015).

The same region of Cyt.*c* interacts with Cyt.*bc*₁ in the Cyt.*bc*₁– Cyt.*c* complex crystal (Lange & Hunte, 2002; Solmaz & Hunte, 2008) and with CcO in the Cyt.*c*–CcO complex crystal, as proposed based on the results of a chemical modification study (Rieder & Bosshard, 1980). Cyt.*c* receives and donates electrons through the same site via a repeated association/dissociation process. The novel mode of protein–protein interaction discovered in this study is likely to decrease the potential barrier caused by structural changes upon association/dissociation of the Cyt.*c*–CcO complex because conformational change of both proteins and rearrangement of surface waters are not required upon docking. Therefore, soft and specific contact between the two proteins is important for efficient donation of four electrons from Cyt.*c* to CcO for the $O_2$ reduction reaction. It is remarkable that the X-ray structure of the interface of Cyt.*c*–CcO complex facilitating the electron transfer from heme *c* to $Cu_A$ is greatly different from that from heme $c_1$ to heme *c*, indicating the electron tranfering structures in these two complexes are specialized for the different electron transfer processes [e.g., two electron transfer from heme $c_1$ to heme *c* versus four electron transfer from heme c to $Cu_A$; and different molar contents of 3:7:9 for Cyt.*bc*₁, CcO, and Cyt.*c* (Hatefi & Galante, 1978)]. Further structural and functional comparisons of these complexes would develop insights in the mechanism of the energy trunsduction by the mitochondrial electron transfer system.

We hypothesize that there are many cases of soft and specific protein–protein interactions involved in various cellular processes. One reason why these interactions were not discovered previously may be related to the need to perform extensive searches for optimal crystallization conditions for these intrinsically unstable protein complexes, as described above. Because single-particle analysis by cryo-electron microscopy does not require a crystal, a high-resolution single-particle analysis would increase the chance of detecting soft and specific protein–protein interaction.

# Materials and Methods

### Preparation of horse heart Cyt.*c* sample

For each crystallization trial, horse heart Cyt.*c* (Nacalai Tesque) was freshly dissolved in 15 mM sodium phosphate buffer at pH 8.0, and then dialyzed for 1 h against the same buffer to remove remaining salts. The concentration of Cyt.*c* was calculated from the absorption spectrum of the fully dithionite-reduced form, using $\Delta\varepsilon_{550-535\ nm} = 19.2\ mM^{-1}\ cm^{-1}$.

### Crystallization of the Cyt.*c*–CcO complex

CcO in the fully oxidized state was purified from bovine heart mitochondria (Tsukihara *et al*, 1995) and dissolved in 40 mM sodium phosphate buffer (pH 6.8) containing 0.2% (w/v) *n*-decyl-β-D-maltoside (DM) (Dojin). CcO was diluted 10-fold in 15 mM sodium phosphate buffer (pH 8.0) containing 0.7% (w/v) fluorinated octyl-maltoside (FOM) (Anatrace). CcO at pH 8.0 preserved Cyt.*c*

oxidation activity at ~50% of the level at pH 7.0, as reflected by $V_{max}$ (Yonetani & Ray, 1965). FOM-treated C*c*O was concentrated using a membrane filter (Amicon Ultra Centrifugal Filters (100 kDa), Millipore). The concentration of C*c*O was calculated from the absorption spectrum of the fully dithionite-reduced form, using $\Delta\varepsilon_{604\text{–}630\ nm} = 46.6\ mM^{-1}\ cm^{-1}$. C*c*O solubilized with DM and FOM was mixed with Cyt.*c* at a Cyt.*c*/C*c*O molar ratio of 1.2. Co-crystallization of Cyt.*c* and C*c*O was performed by the batch-wise method at 277 K; Cyt.*c*–C*c*O (100 mg/ml C*c*O, ~0.5 mM C*c*O) was mixed with ~5% polyethylene glycol (PEG) 4000, a precipitant. Rectangular plates of Cyt.*c*–C*c*O complex crystals were obtained within 1 day. The crystals were gradually soaked in a crystallization solution containing both 50 μM Cyt.*c* and ethylene glycol (EG) as cryo-protectant, reaching final concentrations of 40% EG and 6% PEG 4000. After cryo-protection, crystals were quickly frozen in a cryo-nitrogen stream at 100 K.

## Structure determination

Crystal screening and X-ray experiments were carried out at beam-line BL26B2 and BL44XU of SPring-8. The dataset for the structural analysis was obtained at BL44XU, equipped with an MX300HE CCD detector. The X-ray beam cross-section for X-ray diffraction experiments was $50 \times 50$ μm at the crystal, and the wavelength was 0.9 Å. Photon number at the sample position was $3.0 \times 10^{11}$ photons/s. For data acquisition at 100 K, the crystals were frozen in a cryo-nitrogen stream. The dataset was collected with an exposure time of 1 s and a 0.5° oscillation angle over 180°. Diffraction images were processed and scaled with DENZO and SCALEPACK (Otwinowski & Minor, 1997), respectively, and the datasets from the two crystals were merged. A total of 720 images were successfully processed and scaled. The structure factor amplitude ($|Fo|$) was calculated using the CCP4 program TRUNCATE (French & Wilson, 1978; Weiss, 2001; Winn *et al*, 2011). The crystal belongs to space group $P2_1$, with unit cell dimensions of $a = 113.3$ Å, $b = 183.9$ Å, $c = 148.9$ Å, and $\beta = 102.1°$. The asymmetric unit of the crystal lattice contains two complexes of C*c*O and Cyt.*c*. The solvent content and $V_M$ were 65.6% and 3.58 $Å^3$ $Da^{-1}$, respectively (Matthews, 1968).

C*c*O was initially located in the unit cell at 3.0-Å resolution by the molecular replacement (MR) method (Rossmann & Blow, 1962) using the program MOLREP in CCP4 (Collaborative Computational Project 4, 1994) with the fully oxidized C*c*O structure, previously determined at 1.8-Å resolution (PDB 2DYR) (Shinzawa-Itoh *et al*, 2007), as a model. Cyt.*c* was located following the MR search at 3.0-Å resolution using horse Cyt.*c* (PDB 1HRC) (Bushnell *et al*, 1990) as a model, as in the previous case. Initial phases up to 5.0-Å resolution were calculated with atomic parameters determined by MR and extended to 2.0-Å resolution by density modification (Wang, 1985) coupled with non-crystallographic symmetry averaging (Bricogne, 1974, 1976) using the CCP4 program DM (Cowtan, 1994). The resultant phase angles ($\alpha_{MR/DM}$) were used to calculate the electron-density map (MR/DM map) with Fourier coefficients $|F_o|\ exp(i\alpha_{MR/DM})$ and the anomalous difference electron-density map with Fourier coefficients $(|F_o^+| - |F_o^-|)\ exp[i(\alpha_{MR/DM} - \pi/2)]$, where $|F_o|$ is the observed structure amplitude and $|F_o^+| - |F_o^-|$ is the Bijvoet difference in $|F_o|$. The anomalous difference electron-density map indicated the Fe, Cu, and Zn positions, including the heme

irons of Cyt.*c*. The structural model of Cyt.*c*–C*c*O was built in the MR/DM map. The structure was refined using the program REFMAC (Winn *et al*, 2001; Murshudov *et al*, 2011) at 2.0-Å resolution. Bulk solvent correction and anisotropic scaling of the observed and calculated structure amplitudes and TLS parameters were incorporated into the refinement calculation. The anisotropic temperature factors for iron, copper, and zinc atoms were imposed on the calculated structure factors. Because the two crystallographically independent monomers packed differently in the crystal, each monomer of C*c*O was assigned to a single TLS group in the REFMAC refinement. The quality of the structural refinement was characterized by the $R$ and $R_{free}$ values. $F_o$–$F_c$ maps were calculated with Fourier coefficients $(|F_o| - |F_c|)\ exp(i\alpha_c)$, where $|F_c|$ and $\alpha_c$ are the calculated structure amplitude and phase, respectively, obtained in the structural refinement. Out of 3822 amino-acid residues, 56 residues of C*c*O could not be located in the electron-density maps of the Cyt.*c*–C*c*O complex. A total of 33 residues of C*c*O have multiple conformations. The structure refinement was well converged: $R = 0.167$ and $R_{free} = 0.207$. The root-mean-square deviations (r.m.s.d.) of bond lengths and angles from their ideal values were 0.023 Å and 2.0°, respectively.

Electrostatic potential was calculated separately for Cyt.*c* and subunit II of C*c*O using APBS (Baker *et al*, 2001). Accessible surface area was calculated with the program AREAIMOL (Lee & Richards, 1971) in CCP4, using a probe with radius 1.4 Å.

## Electron transfer pathway calculation

The possible ET pathways from the heme *c* iron to $Cu_A$ of C*c*O were explored by an empirical method, the *Pathways* plugin for VMD (Humphrey *et al*, 1996; Balabin *et al*, 2012), which evaluates the donor-to-acceptor tunneling coupling ($T_{DA}$) value for each pathway. We attached hydrogen atoms based on the charmm22 force field (MacKerell *et al*, 1998), and performed energy minimization with respect to the hydrogen atoms, which enabled us to positively identify the hydrogen bonds. To evaluate the diversity of the possible ET pathways, we generated 200 candidates, and found that most of the identified ET pathways and their $T_{DA}$ values were similar. In fact, the top 100 solutions contained the route between Lys[13] (Cyt.*c*) and Tyr[105] (C*c*O). Accordingly, we took the ET pathway with the most efficient $T_{DA}$ value as the final solution.

## Accession numbers

Atomic coordinates and structure factors have been deposited in the Protein Data Bank with accession code 5IY5.

**Expanded View** for this article is available online.

## Acknowledgements

We thank the staff members of BL26B2/SPring-8 for their extensive support. This work was supported by JST, CREST (to TT), JSPS KAKENHI Grant 26291033 (to SY), and 25287099 (to MS). SY is a senior visiting scientist at the RIKEN Harima Institute. Diffraction data were collected at BL44XU of the SPring-8 facility under proposals 2014A6500, 2014B6500, and 2015A6500, and at BL26B2 under proposals 2014A1846 and 2014A1860. Computations for electron transfer pathway were performed using the Computer Center for Agriculture, Forestry, and Fisheries Research, MAFF, Japan.

    

## Author contributions

KS-I, SY, and TT designed the research; SS, KS-I, JB, SA, AS, EY, JK, MT, SY, and TT performed the research; SS, KS-I, and SA performed protein purification and crystallization experiments; SS, KS-I, JB, SA, AS, EY, and TT performed X-ray diffraction experiments and analyzed X-ray data; JK and MT carried out molecular dynamics simulations; SS, KS-I, AS, MT, SY, and TT wrote the manuscript; and all authors discussed and commented on the results and the manuscript.

## Conflict of interest

The authors declare that they have no conflict of interest.

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
