## [Review Process File · The EMBO Journal]

Manuscript EMBO-2016-95021

Complex structure of cytochrome c–cytochrome c oxidase reveals a novel protein-protein interaction mode

Satoru Shimada, Kyoko Shinzawa-Itoh, Junpei Baba, Shimpei Aoe, Atsuhiko Shimada, Eiki Yamashita, Jiyoung Kang, Masaru Tateno, Shinya Yoshikawa and Tomitake Tsukihara

*Corresponding author: Tomitake Tsukihara, Osaka University;
Kyoko Shinzawa-Itoh, University of Hyogo*

Review timeline:

Submission date:	16 June 2016
Editorial Decision:	05 August 2016
Revision received:	21 October 2016
Accepted:	16 November 2016

Editor: Anne Nielsen

Transaction Report:

1st Editorial Decision

05 August 2016

Thank you for submitting your manuscript for consideration by the EMBO Journal and my apologies for the slightly extended review period. Your study has now been seen by three referees and their comments are included below.

As you will see from the reports, all three referees highlight the interest and value in determining the first co-crystal structure of a mammalian CcO-CytC complex and in presenting the three-layered water interface between the subunits as a new type of protein-protein interaction. However, you will also see that the referees (especially refs #2 and #3) have some suggestions and requests for additional discussions and experiments to be included before they can support publication of the manuscript here. Most importantly, ref #3 finds that more data is required to conclude that the structure presented here corresponding to a functionally and physiologically relevant form. Finally, you will see that the refs are interested in knowing more about the generality and contribution from the soft and specific interaction reported here.

Given the referees' positive recommendations, I would like to invite you to submit a revised version of the manuscript, addressing the comments of all three reviewers. I should add that it is EMBO Journal policy to allow only a single round of revision, and acceptance of your manuscript will therefore depend on the completeness of your responses in this revised version.

When preparing your letter of response to the referees' comments, please bear in mind that this will form part of the Review Process File, and will therefore be available online to the community. For more details on our Transparent Editorial Process, please visit our website: http://emboj.embopress.org/about#Transparent_Process

Thank you for the opportunity to consider your work for publication. I look forward to your revision.

REFeree REPORTS

Referee #1:

Shimada et al. report the crystal structure of the bovine CcO-horse Cyt.c complex at 2.0 Å resolution. CcO accepts electrons from Cyt.c to reduce O₂ to H₂O, and pumps protons across the membrane to create membrane potential. Thus, CcO and Cyt.c play essential roles in mitochondrial energy generation. Although the structure of a bacterial CcO protein, which has a covalently-fused Cyt.c domain, has been reported a few years ago (Lyons et al., 2012), it is not clear whether this bacterial CcO-Cyt.c protein and a eukaryotic CcO-Cyt.c complex function analogously, and thus the structure of a eukaryotic complex is needed to fully understand the electron transfer mechanism. The authors and co-workers previously reported the 2D crystallization of the mammalian Cyt.c-CcO complex (Osuda et al., 2016), but it did not allow the structural determination.

Here, the authors have improved the crystallization conditions and obtained 3D crystals of the mammalian Cyt.c-CcO complex, from which the authors were able to determine its first X-ray structure at high resolution. Important novel findings of this study are (1) the identification of specific interaction between CcO and Cyt.c, (2) the description of a possible electron transfer pathway between Cyt.c and CcO, and (3) the identification of three layers of water molecules at the protein interface, which probably contribute to the complex formation. These findings significantly advance our understanding of the mechanism of electron transfer between the Cyt.c and CcO in eukaryotes. In addition, a new kind of protein-protein contact, which the authors termed as "soft and specific", has a potential general interest in that it can be applied to other transient protein complexes.

Overall, this is a nice study that deserves publication in EMBO Journal. I do not believe that the authors need additional experiments to support their conclusions.

I have the following minor concerns that needs to be addressed:

- 1) In pages 6 to 7, the authors do not discuss the possibility that the Cyt.c binding-site identified in the current structure can be the second (non-catalytic) binding-site. The authors should discuss reasons why this possibility is excluded, or otherwise they should reduce their tone.
- 2) Electron density for Cyt.c seems very poor in the MR/DM map shown in Fig. S7. Given the weak interaction between the components and high B-factor of Cyt.c, I suspect that Cyt.c molecules might be partially occupied or severely disordered. I suggest that the authors present the refined 2Fo-Fc map and the omit Fo-Fc map for Cyt.c to show the quality of their model.
- 3) The location of Trp104 should be presented to support the discussion in the second paragraph of page 7.
- 4) Since the water molecules at the Cyt.c-CcO interface are important in supporting one of the main conclusions, electron density maps of these water molecules should be presented.
- 5) If the water molecules are involved in "specific contact" between the components, it is likely that the hydrogen-bonding network of water molecules is similar between the two molecules (mol A and

B) in the crystal lattice. Thus, superimposition of water molecules (and proteins) in mol A and B should be presented to clarify whether this is the case.

6) Why "soft and specific contact" is not observed in the Cyt.c-Cyt.bc1 complex, which also associates/dissociates repeatedly? The authors should discuss the differences between the two types of complexes, in relation to their modes of action.

7) In line 10 of page 10, "provide flexibility with Cyt.c" seems to be a mistype. It should be corrected as "provide Cyt.c with flexibility".

In addition, the following is non-essential suggestions:

1) The authors could further analyze PDB and other literatures to find out whether there are examples of "soft and specific interactions" in other non-ET proteins, to expand on the generality of this kind of interaction.

2) In the last paragraph, authors note that "soft and specific interactions" have not been discovered because of the technical difficulties in preparing co-crystals of such unstable protein complexes. This interpretation is plausible, but it is also possible that this kind of interaction is specific to the Cyt.c-CcO complex and not widespread in nature. I feel that the authors could justify their interpretation by discussing possible future directions for studying this kind of interaction in other unstable complexes by complementary methods (e.g. single particle Cryo-EM, single molecule FRET, etc.).

Referee #2:

The structure of a complex between mammalian cytochrome oxidase (CcO) and cytochrome c (Cyt.c) has been a long standing challenge in bioenergetics and the authors are to be congratulated on their success in determining the crystal structure at high resolution. The structure explains existing biochemical data on the interaction between the two proteins and provides an entirely plausible electron transfer pathway, effectively dealing with concerns that the observed interaction could be a crystallization artifact (the lack of any lattice contacts involving Cyt.c also supports this view).

Of particular interest, and the main focus of the manuscript, is the nature of the interface between CcO and Cyt.c, which is relatively small in area and consists of three "layers" of waters, with the central layer not making any direct contact with either protein. The authors rationalize the presence of such an unusual arrangement (soft and specific contact) as being required for efficient donation of four electrons from Cyt.c and CcO, by decreasing the energetic barrier to association/dissociation. While this is reasonable, it is not entirely clear why a similar scheme is not used in other Cyt.c complexes such as with Cyt.bc1 and CcP for example, where efficient electron transfer would also be an advantage; perhaps the authors could comment on this. The authors also make no mention of a study of bound water at protein-protein interfaces (Ahmed et al) that confirms the unusual nature of the Cyt.c-CcO interface but also describes how waters at protein-protein interfaces that are involved in limited hydrogen bonding are often associated with hydrophobic side chains. Is that the case in this structure, or is the hydrogen bonding potential of all the waters shown in Figure 4A satisfied?

Finally there is a paper in press in JBC (Sato et al) using protein docking simulations and experimental kinetic data that proposes hydrophobic interactions as the primary factor promoting complex formation between Cyt.c and CcO, some comment on the conclusions of this paper would be valuable.

A number of minor issues are listed below:

1. In several places, distances are quoted to two decimal places which is not justified by the accuracy of the structures, eg 6.98 (p.5), 8.24, 5.28, 5.57 on p. 8.

2. p. 8 I was not entirely convinced by the use of the shortest distance between two Calpha atoms as

being a good way to characterize the nature of the different interfaces involving Cyt.c, as the difference (~3Å) is much shorter than the length of many side chains and could, in principle, simply be the result of different side chains at the interface. The buried surface area is more convincing.

3. p. 8 Regarding the contact surface area calculations, how were water molecules treated in the calculations, were they considered to be part of the protein or ignored?

4. p.9 final paragraph, the B factors quoted are surely average B factors (average is missing).

5. I found the use of the term "second category" on p.10 to describe waters not contacting either protein not very helpful, in that I needed to go back and check what "second category" was, perhaps it could just be spelt out as contacting neither protein?

6. Some references need to be updated:

The CCP4 reference is now:

Winn, M.D., Ballard, C.C., Cowtan, K.D., Dodson, E.J., Emsley, P., Evans, P.R., Keegan, R.M., Krissinel, E.B., Leslie, A.G.W., McCoy, A., McNicholas, S.J., Murshudov, G.N., Pannu, N.S., Potterton, E.A., Powell, H.R., Read, R.J., Vagin, A., Wilson, K.S. 2011. Overview of the CCP4 suite and current developments. *Acta Cryst. D* 67, 235-242.

The proper Refmac reference is:

Murshudov et al., 2011, *Acta Crystallogr D Biol Crystallogr*, 67, 355-67

References

Ahmed, Mostafa H.; Spyarakis, Francesca; Cozzini, Pietro; et al.

Bound Water at Protein-Protein Interfaces: Partners, Roles and Hydrophobic Bubbles as a Conserved Motif. *PLOS ONE* Volume: 6 Issue: 9 Article Number: e24712 Published: SEP 22 2011

Wataru Sato, Seiji Hitaoka, Kaoru Inoue, Mizue Imai, Tomohide Saio, Takeshi Uchida, Kyoko Shinzawa-Itoh, Shinya Yoshikawa, Kazunari Yoshizawa, and Koichiro Ishimori. Energetic Mechanism of Cytochrome c - Cytochrome c Oxidase Electron Transfer Complex Formation under Turnover Conditions Revealed by Mutational Effects and Docking Simulation *J. Biol. Chem.* jbc.M115.708065. doi:10.1074/jbc.M115.708065

Referee #3:

The formation of transient electron transfer complexes is an important common principle in respiratory and photosynthetic energy metabolism. Small diffusible redox proteins facilitate electron transfer by alternately binding to integral membrane proteins. Specific and transient complexes are formed between the redox partners to enable fast turnover numbers. In the mitochondrial respiratory chain, the diffusible one-electron carrier cytochrome c (cyt c) shuttles electrons between complex III (bc1) and complex IV (CcO). So far, a structure of mitochondrial CcO with bound cyt c was lacking. In this study, the authors co-crystallized oxidized horse heart cyt c and oxidized bovine CcO at pH 8.0, and determined the X-ray structure of the complex at 2.0 Å resolution. The structure of the complex suggests a possible intramolecular electron transfer pathway. The interface is highly hydrated and stabilized by electrostatic interactions. The interfacial water molecules are described as "three water layers". The authors term this interaction as "soft and specific", suggesting that this is a new class of protein-protein interaction, which provides the structural basis for the highly transient electron transfer complex. The structural characterization of a CcO - cyt c complex has been long awaited and provides valuable information about the nature of the interface. There is a main point of criticism. It is not clear whether the structure represents a physiological electron transfer complex or whether crystallization at low ionic strength and alkaline pH has trapped an encounter complex or a non-physiological state. Detailed questions and comments are listed below.

CcO activity is highly influenced by ionic strength. Affinity for cyt c is apparently higher at low ionic strength. The authors should precisely specify ionic strength of co-crystallization conditions and of soaking buffer. It seems that the structure was obtained at ionic strength well below the physiological range. This should be included and discussed in the manuscript.

It would be of interest for the reader to include a comment, whether the addition of 50 μM cyt c in the soaking buffer had an effect on the quality of crystals/structure.

The turnover number of CcO depends on pH, it is less active at alkaline pH. What is the activity at pH 8.0? The pH dependence of CcO should be mentioned and discussed in the manuscript.

page 5 - subunit II is highlighted in Fig. 1 and cyt c binding appears to be close to it (although it is difficult to see in this figure, which could be improved showing a section of the structure at the interface). It would be informative to state which subunits contribute to the concave binding site.

Fig. S2 - Figure should be updated- a close up view of A. focused on cyt c and the interface might provide clearer information. Is there Fo-Fc difference density covering CcO?

Fig. S3 is not clearly excluding contacts. Can the authors fully exclude that cyt c binding is affected by the crystal lattice? The authors state that cyt c is not "directly" interacting with any neighboring CcO-cyt c complex with the closest distance of 6.98 Å. Are there any indirect interactions, for instance water-mediated hydrogen bonds, ethylglycol or other molecule-mediated interactions?

RMSD values should be given for superimposition Fig S4.

page 6 - There are two concepts to evaluate electron transfer (ET), one makes use of distinct ET pathways (used here) and the other employs the electron-tunneling model for redox proteins of Dutton and colleagues. For better comparison with other electron transfer complexes and with kinetic data (which should be mentioned), the authors should add ET analysis with the electron-tunneling model. The authors should also include discussion whether the 41.9 Å long ET pathway with two through space jumps would permit physiological ET rates. Evolutionary sequence conservation of ET pathway residues (Lys13 of cyt c, Tyr 105 and Met 207 of SUII) should be discussed (in addition to the comparison with caa3).

Fig. 3 A,B - The dashed-line circles should be explained in the figure legend.

page 8 Polar interactions between cyt c and CcO are described. It should be stated in the manuscript whether all side chain orientations are well defined. A stereo 2Fo-Fc electron density map of the relevant section of the interface area should be included (can be supplementary figure).

page 9 - comparison with cyt c - bc1 and cyt c - peroxidase complexes; the authors state that the cyt c - CcO complex has a smaller interface area and a larger distance between the partners ; noteworthy is furthermore, that cyt c interacts in these complexes (in contrast to the novel structure) via its nonpolar surface that surrounds the heme cleft and weak polar interactions are present at the periphery.

page 9 - a more detailed description of interfacial water molecules should be provided to (H-bonds, B-factors, ligating residues) as they most likely provide a water-mediated H-bond network between the proteins . Water molecules that do not directly interact with either protein must be H-bonded to water molecules of the first shell. Also, none of the figures shows so far where the water molecules are located in respect to the polar side chain interactions.

page 9 "water molecules on the surface of both proteins are retained". It is not clear whether water molecules are retained - for this statement, positions of the interfacial water molecules would have to be compared with those present in CcO and cyt c structures. As the pH values of the crystallization conditions are different, the analysis might not be as informative as needed. Water molecules might also be displaced when ion pairs or hydrogen bonds are formed. "when cyt c docks with ccp or cyt bc1 .. exclusion of water molecules ". These interfaces are also well hydrated (though less than in the CcO-Cyt c complex) favored by limited surface complementarity. The main difference is the close interaction of nonpolar residues. The extent of desolvation for these complexes and its energetic contribution to the binding interaction is not understood in detail so far.

page 10 - water molecules that do not interact with either protein are described as playing a role as a cushion - The polar contacts of the interface have a triangular or trapezoid distribution on the surface of cyt c (Fig. 6). As the long residue side chains are apparently well ordered, these contacts appear to provide a planar rigid interaction of the two molecules (supported by B-factor distribution

in cyt c). Thus, one may argue that the rigid environment provides the structure for the water network, rather than water molecules providing a "cushion in the docking of cyt c to CcO, which provide flexibility with cyt c".

page 10 - "the novel type of protein-protein interaction is likely to decrease the potential barrier to the association/dissociation process of the Cyt c-CcO complex". Though the interface is small, there are 6 rigid ion pair/hydrogen bond interactions, which are not favorable for a transient interaction. It is not clearly described, how the water molecules contribute to the binding interaction.

Cyt c and CcO may form multiple encounter complexes and the authors should consider and discuss that the crystallization conditions at low ionic strength and high pH may have trapped an encounter complex which might not be necessarily the physiological electron transfer complex. Characteristic for other transient electron-transfer complexes (cyt c peroxidase-cyt c, bacterial reaction centre-cytochrome c2, and cyt c -bc1 complexes) is a small interface of hydrophobic interactions surrounded by charged residues (mainly long distance).

Figure 4 - In figure 4A, it seems that a loop of CcO collides with cyt c? Is this a problem with 3D? 3D is correct for panels B and C.

Are there any differences in side chain orientations of CcO residues of the interface area as compared to structures of CcO alone?

1st Revision - authors' response

21 October 2016

First of all, we are very much appreciating to get valuable comments from three referees. While we are not sure whether we could suitably respond to comments by these referees, we are happy to respond three referees' comments and would like to address to their comments one by one as following.

Referee #1:

Shimada et al. report the crystal structure of the bovine CcO-horse Cyt.c complex at 2.0 Å resolution. CcO accepts electrons from Cyt.c to reduce O₂ to H₂O, and pumps protons across the membrane to create membrane potential. Thus, CcO and Cyt.c play essential roles in mitochondrial energy generation. Although the structure of a bacterial CcO protein, which has a covalently-fused Cyt.c domain, has been reported a few years ago (Lyons et al., 2012), it is not clear whether this bacterial CcO-Cyt.c protein and a eukaryotic CcO-Cyt.c complex function analogously, and thus the structure of a eukaryotic complex is needed to fully understand the electron transfer mechanism. The authors and co-workers previously reported the 2D crystallization of the mammalian Cyt.C-CcO complex (Osuda et al., 2016), but it did not allow the structural determination.

Here, the authors have improved the crystallization conditions and obtained 3D crystals of the mammalian Cyt.c-CcO complex, from which the authors were able to determine its first X-ray structure at high resolution. Important novel findings of this study are (1) the identification of specific interaction between CcO and Cyt.c, (2) the description of a possible electron transfer pathway between Cyt.c and CcO, and (3) the identification of three layers of water molecules at the protein interface, which probably contribute to the complex formation. These findings significantly advance our understanding of the mechanism of electron transfer between the Cyt.c and CcO in eukaryotes. In addition, a new kind of protein-protein contact, which the authors termed as "soft and specific", has a potential general interest in that it can be applied to other transient protein complexes.

Overall, this is a nice study that deserves publication in EMBO Journal. I do not believe that the authors need additional experiments to support their conclusions.

I have the following minor concerns that needs to be addressed:

1) In pages 6 to 7, the authors do not discuss the possibility that the Cyt.c binding-site identified in the current structure can be the second (non-catalytic) binding-site. The authors should discuss reasons why this possibility is excluded, or otherwise they should reduce their tone.

Response: “Novel protein-protein interaction scheme” of the previous manuscript was divided into two parts, “Catalytic binding sites” and “Novel protein-protein interaction scheme” to discuss the catalytic binding sites. The binding site on CcO for Cyt.c was discussed by following two models, a two catalytic site model and a single catalytic site model that have been proposed previously. Following paragraph was added in page 8; “Speck et al (1984) proposed a single catalytic site model including a catalytic site and a noncatalytic regulatory site on CcO for Cyt.c to interpret the steady state kinetic results indicating two different Michaelis-Menten kinetics, without giving any experimental confirmation. In other words, no experimental result has disproven the two catalytic site model (Ferguson-Miller et al, 1976). Following the Speck’s definition, the above structure strongly suggests the catalytic binding site, since the Cyt.c-CcO complex shows a facile electron transfer pathway from heme c to Cu_A. However, following the two catalytic site model, this retains both possibilities of the first and the second catalytic sites.”

2) Electron density for Cyt.c seems very poor in the MR/DM map shown in Fig. S7. Given the weak interaction between the components and high B-factor of Cyt.c, I suspect that Cyt.c molecules might be partially occupied or severely disordered. I suggest that the authors present the refined 2Fo-Fc map and the omit Fo-Fc map for Cyt.c to show the quality of their model.

Response: According to this comment Fo-Fc map for Cyt.c, Fo-Fc map for heme c, anomalous difference map for Fe of the heme c and 2Fo-Fc map for Cyt.c-CcO interface are given in Fig S2. Poor electron density for Cyt.c is implicated as a large fluctuation using the regions interacting with CcO as a fulcrum. Side chains interacting with CcO whose B-factors are as low as those of CcO atoms are located in clear shapes of electron density.

3) The location of Trp104 should be presented to support the discussion in the second paragraph of page 7.

Response: The location of Trp104 is shown in Fig S7, a stereoscopic drawing of the region including heme c of Cyt.c and Cu_A of CcO.

4) Since the water molecules at the Cyt.c-CcO interface are important in supporting one of the main conclusions, electron density maps of these water molecules should be presented.

Response: According to the referee’s suggestion a 2Fo-Fc map for these water molecules is given in EV 4A.

5) If the water molecules are involved in "specific contact" between the components, it is likely that the hydrogen-bonding network of water molecules is similar between the two molecules (mol A and B) in the crystal lattice. Thus, superimposition of water molecules (and proteins) in mol A and B should be presented to clarify whether this is the case.

Response: The second paragraph of “**Structure determination and overall structure**”, was added in page 5 to explain the relationship of two molecules A and B as follows; “As in the CcO orthorhombic crystal (Tomizaki et al, 1999), one of the two CcO molecules in the asymmetric unit had a lower B-factor than the other, by about 7 Å², and no significant structural difference was detected between the two complexes. Furthermore structure refinement was performed under non-crystallographic symmetry restraint between two CcO molecules; therefore, we focused our structural descriptions on this complex.” For example, out of 64 water molecules that are located between Cyt.c and CcO of A, 51 have equivalent water molecules in the B molecule, while each of the remaining 13 equivalent sites has a space for a water but its electron density is too low to locate a water molecule.

6) Why "soft and specific contact" is not observed in the Cyt.c-Cyt.bc1 complex, which also associates/dissociates repeatedly? The authors should discuss the differences between the two types of complexes, in relation to their modes of action.

Response: In page 14, following sentences are added to discuss it; “It is remarkable that the X-ray structure of the interface of Cyt.c-CcO complex facilitating the electron transfer from heme c to Cu_A is greatly different from that from heme c₁ to heme c, indicating the electron transferring structures in these two complexes are specialized for the different electron transfer processes (for example, two electron transfer from heme c₁ to heme c vs four electron transfer from heme c to Cu_A; and different molar contents of 3 : 7 : 9 for Cyt.bc₁, CcO and Cyt.c (Hatefi & Galante 1978)). Further structural and functional comparisons of these complexes would develop insights in the mechanism of the energy transduction by the mitochondrial electron transfer system.”

7) In line 10 of page 10, "provide flexibility with Cyt.c" seems to be a mistype. It should be corrected as "provide Cyt.c with flexibility".

Response: It was revised as was pointed out by the reviewer.

In addition, the following is non-essential suggestions:

1) The authors could further analyze PDB and other literatures to find out whether there are examples of "soft and specific interactions" in other non-ET proteins, to expand on the generality of this kind of interaction.

Response: Inter molecular Ca-Ca distances were calculated for 179 protein-protein complexes compiled by Ahmed et al. from PDB. The statistics of inter molecular Ca-Ca distances are given in Fig S9.

2) In the last paragraph, authors note that "soft and specific interactions" have not been discovered because of the technical difficulties in preparing co-crystals of such unstable protein complexes. This interpretation is plausible, but it is also possible that this kind of interaction is specific to the Cyt.c-CcO complex and not widespread in nature. I feel that the authors could justify their interpretation by discussing possible future directions for studying this kind of interaction in other unstable complexes by complementary methods (e.g. single particle Cryo-EM, single molecule FRET, etc.).

Response: As the final sentence of "Novel protein-protein interaction scheme", was added in page 15; "Because single-particle analysis by cryo-electron microscopy does not require a crystal, a high-resolution single-particle analysis would increase the chance of detecting soft and specific protein-protein interaction."

Referee #2:

The structure of a complex between mammalian cytochrome oxidase (CcO) and cytochrome c (Cyt.c) has been a long standing challenge in bioenergetics and the authors are to be congratulated on their success in determining the crystal structure at high resolution. The structure explains existing biochemical data on the interaction between the two proteins and provides an entirely plausible electron transfer pathway, effectively dealing with concerns that the observed interaction could be a crystallization artifact (the lack of any lattice contacts involving Cyt.c also supports this view).

Of particular interest, and the main focus of the manuscript, is the nature of the interface between CcO and Cyt.c, which is relatively small in area and consists of three "layers" of waters, with the central layer not making any direct contact with either protein. The authors rationalize the presence of such an unusual arrangement (soft and specific contact) as being required for efficient donation of four electrons from Cyt.c and CcO, by decreasing the energetic barrier to association/dissociation. While this is reasonable, it is not entirely clear why a similar scheme is not used in other Cyt.c complexes such as with Cyt.bc1 and CcP for example, where efficient electron transfer would also be an advantage; perhaps the authors could comment on this.

Response: In page 14, following sentences are added to discuss it; "It is remarkable that the X-ray structure of the interface of Cyt.c-CcO complex facilitating the electron transfer from heme c to CuA is greatly different from that from heme c₁ to heme c, indicating the electron transferring structures in these two complexes are specialized for the different electron transfer processes (for example, two electron transfer from heme c₁ to heme c vs four electron transfer from heme c to CuA; and different molar contents of 3 : 7 : 9 for Cyt.bc₁, CcO and Cyt.c (Hatefi & Galante 1978)). Further structural and functional comparisons of these complexes would develop insights in the mechanism of the energy transduction by the mitochondrial electron transfer system."

The authors also make no mention of a study of bound water at protein-protein interfaces (Ahmed et al) that confirms the unusual nature of the Cyt.c-CcO interface but also describes how waters at protein-protein interfaces that are involved in limited hydrogen bonding are often associated with hydrophobic side chains. Is that the case in this structure, or is the hydrogen bonding potential of all the waters shown in Figure 4A satisfied?

Response: Ahmed et al. assigned the interface water molecules, which are within 4Å of both proteins, and categorize them into three groups. In the present study the interface water molecules

within 7Å of both proteins of Cyt.c and CcO fall into three categories. When the water molecules within 4Å of both proteins of Cyt.c and CcO are assigned as the interface water molecules, more than 60% of water molecules existing between two proteins are excluded from the interface water molecules.

Inter-molecular Ca-Ca distances for 179 protein-protein complex structures compiled by Ahmed et al. were calculated and added following sentences in page 10; “Ahmed et al (2011) compiled 179 X-ray structures of protein-protein complexes from the RSCB Protein Data Bank (Berman et al, 2000). The inter-molecular Ca distances of these 179 structures were calculated, and the distribution of the shortest distance in each complex is illustrated in Appendix Fig S9. Notably, the shortest distance in the Cyt.c–CcO complex, 8.2 Å, falls well outside the distribution.”

Atoms and amino acids that interact with the water molecules are inspected and their statistics are given in Tables S2 and S3. Hydrogen bonds for the water molecules are shown in EV4 in part. The sentences “Any water in the Cyt.c–CcO complex has at least one hydrogen bond with a protein atom or a water molecule. Each of the bridging and non-bridging waters in the Cyt.c–CcO complex interacts, on average, with three polar atoms or waters and one non-polar atom (Table S2). The waters at CcO interact prominently with Asp, whereas those at Cyt.c interact mainly with Lys and Gln (Table S3). At least two water molecules closely contact a non-interacting water molecule. These water molecules construct a hydrogen bond network between Cyt.c and CcO (Fig EV4).” were added in pages 11-12.

Finally there is a paper in press in JBC (Sato et al) using protein docking simulations and experimental kinetic data that proposes hydrophobic interactions as the primary factor promoting complex formation between Cyt.c and CcO, some comment on the conclusions of this paper would be valuable.

Response: The interacting residues of Cyt.c proposed in this study are quite consistent with the kinetic experiments coupled with site-directed mutagenesis by Sato et al., whereas their proposal on the primary factor of complex formation conflicts with the present structure in part. These are explained by following sentences; “Recent site-directed mutagenesis and kinetics studies of Cyt.c indicated that the ET activities of K13L, K86L/K87L, and K7L/K8L mutants are significantly lower than that of the wild-type protein (Sato et al, 2016). The side chains of Lys⁸, Gln¹², Lys¹³, and Lys⁸⁷ of Cyt.c, as well as the side chains of Tyr¹⁰⁵, Asp¹¹⁹, Ser¹¹⁷, Tyr¹²¹, and Asp¹³⁹ of CcO subunit II, provide the physiological electron transfer complex, not an encounter complex under non-physiological condition.” in pages 8-9; and “By contrast, our crystal structure of the Cyt.c–CcO complex has no intermolecular interactions between hydrophobic amino acids with an inter-atomic distance less than 5 Å. This is likely because NMR spectroscopy sensitively detected a small structural change undetectable by X-ray, mediated by an interaction between the residues of Cyt.c and CcO via water molecules present between the two proteins.” in page 9.

A number of minor issues are listed below:

1. In several places, distances are quoted to two decimal places which is not justified by the accuracy of the structures, eg 6.98 (p.5), 8.24, 5.28, 5.57 on p. 8.

Response: As DPI estimate of positional s of the present structure is ~0.1 or more, these figures were given to one decimal places as suggested by the referee.

2. p. 8 I was not entirely convinced by the use of the shortest distance between two Calpha atoms as being a good way to characterize the nature of the different interfaces involving Cyt.c, as the difference (~3Å) is much shorter than the length of many side chains and could, in principle, simply be the result of different side chains at the interface. The buried surface area is more convincing.

Response: The shortest Ca-Ca distance that can be estimated for lower resolution structure and more easily than the case of buried surface is an efficient criterion for protein-protein interaction.

3. p. 8 Regarding the contact surface area calculations, how were water molecules treated in the calculations, were they considered to be part of the protein or ignored?

Response: The sentence explains it in page 10; “Furthermore, the contact surface areas for three ET complexes were calculated by removing surface water molecules.”

4. p.9 final paragraph, the B factors quoted are surely average B factors (average is missing).

Response: The sentence was revised as follows in page 13; “The B-factors of the side-chain atoms of Nz (Lys⁸), Ne (Gln¹²), Nz (Lys¹³), and Nz (Lys⁸⁷) of Cyt.c, which interact with residues of CcO,

are 37.7, 39.3, 41.7, and 48.7 Å², respectively, significantly lower than the average *B*-factor of Cyt.c (88.2 Å²) and as low as that of the extracellular domain (residues 91–227) of CcO subunit II (35.2 Å²).”

5. I found the use of the term "second category" on p.10 to describe waters not contacting either protein not very helpful, in that I needed to go back and check what "second category" was, perhaps it could just be spelt out as contacting neither protein?

Response: These terms were replaced by bridging water, non-bridging water and non-interacting water,

6. Some references need to be updated:

The CCP4 reference is now:

Winn, M.D., Ballard, C.C., Cowtan, K.D., Dodson, E.J., Emsley, P., Evans, P.R., Keegan, R.M., Krissinel, E.B., Leslie, A.G.W., McCoy, A., McNicholas, S.J., Murshudov, G.N., Pannu, N.S., Potterton, E.A., Powell, H.R., Read, R.J., Vagin, A., Wilson, K.S. 2011. Overview of the CCP4 suite and current developments. *Acta Cryst. D67*, 235-242.

The proper Refmac reference is:

Murshudov et al., 2011, *Acta Crystallogr D Biol Crystallogr*, 67, 355-67

Response: The references were revised according to the referee's suggestion.

References

Ahmed, Mostafa H.; Spyraakis, Francesca; Cozzini, Pietro; et al.

Bound Water at Protein-Protein Interfaces: Partners, Roles and Hydrophobic Bubbles as a Conserved Motif. *PLOS ONE* Volume: 6 Issue: 9 Article Number: e24712 Published: SEP 22 2011

Wataru Sato, Seiji Hitaoka, Kaoru Inoue, Mizue Imai, Tomohide Saio, Takeshi Uchida, Kyoko Shinzawa-Itoh, Shinya Yoshikawa, Kazunari Yoshizawa, and Koichiro Ishimori. Energetic Mechanism of Cytochrome c - Cytochrome c Oxidase Electron Transfer Complex Formation under Turnover Conditions Revealed by Mutational Effects and Docking Simulation *J. Biol. Chem.* jbc.M115.708065. doi:10.1074/jbc.M115.708065

Response: These references were added according to the referee's suggestion.

Referee #3:

The formation of transient electron transfer complexes is an important common principle in respiratory and photosynthetic energy metabolism. Small diffusible redox proteins facilitate electron transfer by alternately binding to integral membrane proteins. Specific and transient complexes are formed between the redox partners to enable fast turnover numbers. In the mitochondrial respiratory chain, the diffusible one-electron carrier cytochrome c (cyt c) shuttles electrons between complex III (bc₁) and complex IV (CcO). So far, a structure of mitochondrial CcO with bound cyt c was lacking. In this study, the authors co-crystallized oxidized horse heart cyt c and oxidized bovine CcO at pH 8.0, and determined the X-ray structure of the complex at 2.0 Å resolution. The structure of the complex suggests a possible intramolecular electron transfer pathway. The interface is highly hydrated and stabilized by electrostatic interactions. The interfacial water molecules are described as "three water layers". The authors term this interaction as "soft and specific", suggesting that this is a new class of protein-protein interaction, which provides the structural basis for the highly transient electron transfer complex. The structural characterization of a CcO - cyt c complex has been long awaited and provides valuable information about the nature of the interface. There is a main point of criticism. It is not clear whether the structure represents a physiological electron transfer complex or whether crystallization at low ionic strength and alkaline pH has trapped an encounter complex or a non-physiological state. Detailed questions and comments are listed below.

Response: It is impossible to attain the precise physiological conditions into any crystalline mitochondrial transmembrane protein solubilized from the mitochondrial inner membrane. For example, under physiological conditions, the protein is exposed to two different pH conditions depending on the sidedness. The detergent used for solubilization would be deteriorative to the protein moiety. However, under any condition in which the protein shows its physiological function, the purified sample is highly likely to preserve its native structures.

In addition to chemical modification and kinetic studies by Ferguson-Miller et al., site-directed mutagenesis and kinetic studies by Sato et al. (2016) was taken into account to explain that the present structure is consistent with that of physiological state. Following sentences; “Recent site-directed mutagenesis and kinetics studies of Cyt.c indicated that the ET activities of K13L, K86L/K87L, and K7L/K8L mutants are significantly lower than that of the wild-type protein (Sato et al, 2016). The side chains of Lys⁸, Gln¹², Lys¹³, and Lys⁸⁷ of Cyt.c, as well as the side chains of Tyr¹⁰⁵, Asp¹¹⁹, Ser¹¹⁷, Tyr¹²¹, and Asp¹³⁹ of CcO subunit II, provide the physiological electron transfer complex, not an encounter complex under non-physiological condition.” were added in pages 8-9.

CcO activity is highly influenced by ionic strength. Affinity for cyt c is apparently higher at low ionic strength. The authors should precisely specify ionic strength of co-crystallization conditions and of soaking buffer. It seems that the structure was obtained at ionic strength well below the physiological range. This should be included and discussed in the manuscript.

Response: Crystallization conditions are described in “Crystallization of the Cyt.c-CcO complex” in page 16. It is confirmed by several experiments that the obtained structure is identical to that of physiological state as described before.

It would be of interest for the reader to include a comment, whether the addition of 50 μ M cyt c in the soaking buffer had an effect on the quality of crystals/structure.

Response: A sentence was inserted in page 5; “The addition of 50 mM Cyt.c prevented the crystal from deterioration due to release of Cyt.c molecules from the complex during soaking.”

The turnover number of CcO depends on pH, it is less active at alkaline pH. What is the activity at pH 8.0? The pH dependence of CcO should be mentioned and discussed in the manuscript.

Response: In the respect to crystallization condition at pH 8.0 following sentence was inserted; “CcO at pH 8.0 preserved Cyt.c oxidation activity at ~50% of the level at pH 7.0, as reflected by V_{max} (Yonetani and Ray, 1965).” in page 15.

page 5 - subunit II is highlighted in Fig. 1 and cyt c binding appears to be close to it (although it is difficult to see in this figure, which could be improved showing a section of the structure at the interface). It would be informative to state which subunits contribute to the concave binding site.

Response: In page 6 a sentence was inserted; “CcO interacts with Cyt.c mainly via subunit II, with 94% of the contact surface of CcO with Cyt.c belonging to subunit II, and 5% and 1% of it belonging to subunits VIb and I, respectively.”

Fig. S2 - Figure should be updated- a close up view of A. focused on cyt c and the interface might provide clearer information. Is there Fo-Fc difference density covering CcO?

Response: Figure was updated according to the referee suggestion.

Fig. S3 is not clearly excluding contacts. Can the authors fully exclude that cyt c binding is affected by the crystal lattice? The authors state that cyt c is not "directly" interacting with any neighboring CcO-cyt c complex with the closest distance of 6.98 Å. Are there any indirect interactions, for instance water-mediated hydrogen bonds, ethylenglycol or other molecule-mediated interactions?

Response: Any significant residual structure was not detected at the closest pair.

RMSD values should be given for superimposition Fig S4.

Response: A sentence was inserted in page 12; “The main chain folds of CcO and Cyt.c in Cyt.c-CcO complex are almost identical to those of the corresponding crystals, with Ca r.m.s.d. values of 0.47 Å and 0.41 Å, respectively.”

page 6 - There are two concepts to evaluate electron transfer (ET), one makes use of distinct ET pathways (used here) and the other employs the electron-tunneling model for redox proteins of Dutton and colleagues. For better comparison with other electron transfer complexes and with kinetic data (which should be mentioned), the authors should add ET analysis with the electron-tunneling model. The authors should also include discussion whether the 41.9 Å long ET pathway with two through space jumps would permit physiological ET rates. Evolutionary sequence conservation of ET pathway residues (Lys13 of cyt c, Tyr 105 and Met 207 of SUII) should be discussed (in addition to the comparison with caa3).

Response: The reviewer asked for the additional calculations using another empirical model of electron transfer (involving the tunneling effects). However, our calculations that were described in the original manuscript also involve the evaluations of the tunnel effects (see the cited reference), and do not include any ambiguities in the resultant data, as described in the original version of the manuscript. Accordingly, we do not believe that the further calculations that are within the “identical theoretical levels” as the Reviewer mentioned do make sense and are necessary. Thus, we decided not to perform the further empirical ET calculations in the revised version of the manuscript.

Since the proposed one is the most efficient pathway and is acceptable for the slow electron transfer from Cyt.c to CcO (400 s^{-1} , Michel & Bosshard, H. R. (1989) *Biochemistry* **28**, 244–252), we conclude that any additional description required by the referee is not necessary.

In respect to evolutionary conservation, following sentence was added in page 7; “All vertebrate Cyt.c proteins contain Lys at the 13th residue and Cys at the 14th residue (Appendix Fig S6A); in addition, Tyr105 and Met207 of CcO subunit II are conserved among vertebrates (Appendix Fig S6B).”

Fig. 3 A,B - The dashed-line circles should be explained in the figure legend.

Response: The figure legend was revised according to the referee’s suggestion.

page 8 Polar interactions between cyt c and CcO are described. It should be stated in the manuscript whether all side chain orientations are well defined. A stereo 2Fo-Fc electron density map of the relevant section of the interface area should be included (can be supplementary figure).

Response: Stereo scopic (2Fo-Fc) maps of the interface are given in EV 4A and Fig S2C. To explain structural soundness following sentence was inserted in pages 5-6; “Although B-factors of Cyt.c significantly higher than those of CcO, all the side chains except for Lys²⁵ were located in the positive density of (2Fo-Fc) map.”

page 9 - comparison with cyt c - bc1 and cyt c - peroxidase complexes; the authors state that the cyt c - CcO complex has a smaller interface area and a larger distance between the partners ; noteworthy is furthermore, that cyt c interacts in these complexes (in contrast to the novel structure) via its nonpolar surface that surrounds the heme cleft and weak polar interactions are present at the periphery.

Response: Any direct interaction ($<5.0 \text{ \AA}$) between hydrophobic residues was not in Cyt.c and CcO interface. It is described in page 11 as follows; “No direct interaction ($<5.0 \text{ \AA}$) between hydrophobic residues was detected in the Cyt.c–CcO complex, whereas the other two complexes have several non-polar groups involved in their inter-molecular interactions.”

page 9 - a more detailed description of interfacial water molecules should be provided to (H-bonds, B-factors, ligating residues) as they most likely provide a water-mediated H-bond network between the proteins . Water molecules that do not directly interact with either protein must be H-bonded to water molecules of the first shell. Also, none of the figures shows so far where the water molecules are located in respect to the polar side chain interactions.

Response: EV 4B was prepared to show hydrogen bond network consisting of waters around polar side chains. Tables S1 and S2 shows structural features of water molecules. B-factors of water molecules are higher than that of CcO and lower than Cyt.c. Following paragraph was inserted in pages 11-12; “Any water in the Cyt.c–CcO complex has at least one hydrogen bond with a protein atom or a water molecule. Each of the bridging and non-bridging waters in the Cyt.c–CcO complex interacts, on average, with three polar atoms or waters and one non-polar atom (Table S2). The waters at CcO interact prominently with Asp, whereas those at Cyt.c interact mainly with Lys and Gln (Table S3). At least two water molecules closely contact a non-interacting water molecule. These water molecules construct a hydrogen bond network between Cyt.c and CcO (Fig EV4). The averaged B-factor of 64 waters is 62.7 \AA^2 , which is between that of CcO (37.9 \AA^2) and that of Cyt.c (88.2 \AA^2).”

page 9 "water molecules on the surface of both proteins are retained". It is not clear whether water molecules are retained - for this statement, positions of the interfacial water molecules would have to be compared with those present in CcO and cyt c structures. As the pH values of the crystallization conditions are different, the analysis might not be as informative as needed. Water

molecules might also be displaced when ion pairs or hydrogen bonds are formed. "when cyt c docks with ccp or cyt bc1 .. exclusion of water molecules ". These interfaces are also well hydrated (though less than in the CcO-Cyt c complex) favored by limited surface complementarity. The main difference is the close interaction of nonpolar residues. The extent of desolvation for these complexes and its energetic contribution to the binding interaction is not understood in detail so far.

Response: Non-bridging water sites in the Cyt.c-CcO crystal and water sites in the CcO crystallized at pH6.8 were compared. Most of non-bridging sites were shared with both crystals crystallized in different pH. A paragraph was inserted in page 12 as follows; " Out of 19 non-bridging waters at CcO in the complex, 14 are located at almost the same sites in the CcO crystal (pdb ID, 5B1A), four are in slightly shifted positions, and one water is not assigned in the CcO crystal. Out of 23 non-bridging waters at Cyt.c in the complex, only five waters are present in the Cyt.c crystal structure (pdbID, 1HRC). This is probably because the interacting sites of Cyt.c in the Cyt.c-CcO complex are involved in the tight contacts of crystal packing in the Cyt.c crystal, which removes waters from the molecular surface upon crystallization. Because the protein volumes of Cyt.c-CcO and CcO crystals are ~30% of their unit cell volume, significantly lower than that of Cyt.c crystal, more than 40%, non-bridging water sites are common to the Cyt.c-CcO and CcO crystals."

page 10 - water molecules that do not interact with either protein are described as playing a role as a cushion - The polar contacts of the interface have a triangular or trapezoid distribution on the surface of cyt c (Fig. 6). As the long residue side chains are apparently well ordered, these contacts appear to provide a planar rigid interaction of the two molecules (supported by B-factor distribution in cyt c). Thus, one may argue that the rigid environment provides the structure for the water network, rather than water molecules providing a "cushion in the docking of cyt c to CcO, which provide flexibility with cyt c".

Response: Averaged B-factors of four groups of waters were calculated to inspect their mobility. That of non-interacting waters (66.4 \AA^2) was higher than those of non-bridging waters at CcO (54.9 \AA^2) and bridging waters (58.0 \AA^2), but was not higher than that of non-bridging waters at Cyt.c (68.1 \AA^2). Since non-interacting waters did not exhibit higher mobility than non-bridging water at Cyt.c, they were not provided with a role of cushion in this paper. Thus a sentence in page 14 was revised as follows; "The non-interacting water molecules in the Cyt.c-CcO complex exist in vacant spaces around the interacting amino-acid residues of both proteins (Fig 6) and closely contact with water molecules, thus providing hydrogen bond network between Cyt.c and CcO (EV 4). "

page 10 - "the novel type of protein-protein interaction is likely to decrease the potential barrier to the association/dissociation process of the Cyt c-CcO complex". Though the interface is small, there are 6 rigid ion pair/hydrogen bond interactions, which are not favorable for a transient interaction. It is not clearly described, how the water molecules contribute to the binding interaction.

Response: EV 4 was prepared to show a hydrogen bond network of water molecules and bridge Cyt.c and CcO. Although direct interaction between Cyt.c and CcO is very restrictive, two proteins interact via the hydrogen bond network.

Cyt c and CcO may form multiple encounter complexes and the authors should consider and discuss that the crystallization conditions at low ionic strength and high pH may have trapped an encounter complex which might not be necessarily the physiological electron transfer complex. Characteristic for other transient electron-transfer complexes (cyt c peroxidase-cyt c, bacterial reaction centre-cytochrome c2, and cyt c -bc1 complexes) is a small interface of hydrophobic interactions surrounded by charged residues (mainly long distance).

Response: Whether the present complex is equivalent to the physiologically active one or not was discussed in "Catalytic binding sites" in pages 8-9 and we concluded as follows; "The side chains of Lys⁸, Gln¹², Lys¹³, and Lys⁸⁷ of Cyt.c, as well as the side chains of Tyr¹⁰⁵, Asp¹¹⁹, Ser¹¹⁷, Tyr¹²¹, and Asp¹³⁹ of CcO subunit II, provide the physiological electron transfer complex, not an encounter complex under non-physiological condition. "

Figure 4 - In figure 4A, it seems that a loop of CcO collides with cyt c? Is this a problem with 3D? 3D is correct for panels B and C.

Response: The figure was recast.

Are there any differences in side chain orientations of CcO residues of the interface area as compared to structures of CcO alone?

Response: The side chain orientateon of CcO of the interface area have the same orientation as CcO alone except one amino acid as described in the sentence in page 12; All the CcO side chain structures in the region interacting with Cyt.c are similar to those of the CcO crystal except for Asn203(II), where the two structures are different from each other by a -90° rotation angle around the Cb–Cg bond.”

2nd Editorial Decision

16 November 2016

Thank you for submitting a revised version of your manuscript to The EMBO Journal. It has now been seen by two of the original referees and their comments are shown below. As you will see they both find that all criticisms have been sufficiently addressed and I am therefore happy to inform you that your manuscript has been accepted for publication here.

REFEREE REPORTS

Referee #2:

The revised manuscript suitably addresses the points raised in my original report and is now suitable for publication in EMBOJ

Referee #3:

The authors have adequately addressed questions and comments.

Corresponding Author Name: Kyoko Shinzawa-Itoh and Tomitake Tsukihara

Journal Submitted to: EMBO J

Manuscript Number:EMBOJ-2016-95021